# Deep Learning-Based Monocular 3D Object Detection with Refinement of Depth Information

**DOI:** 10.3390/s22072576

**Published:** 2022-03-28

**Authors:** Henan Hu, Ming Zhu, Muyu Li, Kwok-Leung Chan

**Affiliations:** 1Changchun Institute of Optics, Fine Mechanics and Physics, Chinese Academy of Sciences, Changchun 130033, China; huhenan18@mails.ucas.ac.cn (H.H.); zhu_mingca@163.com (M.Z.); 2University of Chinese Academy of Sciences, Beijing 100049, China; 3Department of Electrical Engineering, City University of Hong Kong, Hong Kong, China; 4Centre for Intelligent Multidimensional Data Analysis Limited, Hong Kong, China; muyu@innocimda.com

**Keywords:** 3D object detection, monocular image, point cloud, deep learning, depth estimation, autonomous driving

## Abstract

Recently, the research on monocular 3D target detection based on pseudo-LiDAR data has made some progress. In contrast to LiDAR-based algorithms, the robustness of pseudo-LiDAR methods is still inferior. After conducting in-depth experiments, we realized that the main limitations are due to the inaccuracy of the target position and the uncertainty in the depth distribution of the foreground target. These two problems arise from the inaccurate depth estimation. To deal with the aforementioned problems, we propose two innovative solutions. The first is a novel method based on joint image segmentation and geometric constraints, used to predict the target depth and provide the depth prediction confidence measure. The predicted target depth is fused with the overall depth of the scene and results in the optimal target position. For the second, we utilize the target scale, normalized with the Gaussian function, as a priori information. The uncertainty of depth distribution, which can be visualized as long-tail noise, is reduced. With the refined depth information, we convert the optimized depth map into the point cloud representation, called a pseudo-LiDAR point cloud. Finally, we input the pseudo-LiDAR point cloud to the LiDAR-based algorithm to detect the 3D target. We conducted extensive experiments on the challenging KITTI dataset. The results demonstrate that our proposed framework outperforms various state-of-the-art methods by more than 12.37% and 5.34% on the easy and hard settings of the KITTI validation subset, respectively. On the KITTI test set, our framework also outperformed state-of-the-art methods by 5.1% and 1.76% on the easy and hard settings, respectively.

## 1. Introduction

Object detection in two-dimensional (2D) images is a classic image processing problem. With the recent advances in deep learning, many deep learning-based 2D object detection models have been proposed. There are generally two types of deep learning-based object detection—one-stage and two-stage [1]. A two-stage detector can achieve high accuracy at the expense of a high computational load. A one-stage detector has faster speed with a little lower accuracy. For some applications, it is better to detect objects in three-dimensional (3D) space. Due to the absence of depth cues in 2D image, monocular 3D object detection is challenging research.

3D object detection methods estimate the 3D-oriented bounding box for each object. The current research on point cloud-based 3D detection algorithms, with input from light detection and ranging (LiDAR) or depth cameras (RGB-D), has made great progress. Alternatively, monocular 3D object detection systems achieve the aforementioned goal with the input of a single 2D image. LiDAR-based methods are fast and accurate. However, the systems are expensive and complicated to set up. Monocular 3D object detection methods, depending on the number of 3D points estimated, are slower. As the main piece of equipment is a video camera, the systems are of low-cost and easy to set up. Similarly to the categorization of 2D object detection models, Kim and Hwang [2] also grouped monocular 3D object detection systems into two main categories—the multi-stage approach and the end-to-end approach. Image-based object detection has gained applications in many fields, such as autonomous driving and robot vision. Intelligent transportation systems demand the localization of pedestrians and vehicles. Chen et al. [3] presented a detailed analysis of many deep learning-based frameworks for pedestrian and vehicle detection. Arnold et al. [4] presented the first survey on 3D object detection for autonomous driving.

Acquiring 3D object bounding boxes from monocular image is an ill-posed problem because 2D images lack the depth information of the objects, which is essential for object localization in 3D space. To solve this missing depth problem, a natural solution [5] is to use convolutional neural networks (CNNs) to regress the depth information. The estimated depth map, with one more dimension, provides additional information for 3D detection. This is referred as the pseudo-LiDAR approach. Although depth estimation is helpful for 3D target detection, the capabilities of monocular depth estimation algorithms is quite limited.

Moreover, a recent study [6] concluded that inaccurate depth estimation is not the only reason for the low accuracy of monocular 3D target detection. It is also due to the fact that the depth map is not suitable for 3D problems. Following this argument, other methods [7,8] were proposed to convert the depth map into a pseudo-point cloud representation. With this 3D data input, a point cloud-based approach can be adopted for 3D detection. Significant performance improvements on the KITTI dataset [9] can be achieved. However, there is still a big gap between the performances of pseudo-LiDAR-based methods and ones based on real 3D point clouds. We conducted thorough experiments and observed that this gap is due to the fact that even high-ranking depth estimation algorithms produce depth maps in which the depths of the foreground objects are still extremely inaccurate. To address this issue, we therefore, focused more on improving the depth accuracy of foreground objects in the depth map. Figure 1a shows the deviation between the pseudo-LiDAR point cloud of a car estimated by the monocular depth estimation method DORN [5] and the ground truth location (the green box). With our proposed depth refinement, as shown in Figure 1b, the pseudo-LiDAR point cloud of the car can be dragged back to the rightful location. Figure 1 demonstrates the result of our depth distribution adjustment. Our proposed method is able to fix the long-tail problem, as shown in Figure 1a,b.

Prior knowledge of an object’s physical size, scene information and the the imaging process of the camera are crucial for depth recovery. Such geometric priors have been exploited to predict objects’ poses or distances. Mono RCNN [10] and GUPNet [11] were proposed. They use the geometric constraint relationship between the physical height of the target and the projected visual height to predict the position of the target more accurately. However, the accuracy of these algorithms is severely affected by the inferred physical height of the target and the pixel height of the target in the image. They suffer from a large number of missed detections of distant targets (more than 50 m). Figure 2 shows a comparison of depth estimation errors (deviations of depth from the center of the object) among the state-of-the-art monocular depth estimation method [5], geometric-constraint-based method [11] and ground truth on the KITTI validation (val) split. DORN [5] can estimate the depths of objects nearby and far away. However, the error increases progressively with the distance of the target. GUPNet [11] can estimate the depths of near targets with reasonably little error. However, the method fails when objects are far away, specifically further than 50 m. As shown in Figure 2, there are no detected depth data at distances beyond 50 m. Other algorithms [7,8] were proposed to solve this problem by forcing deep neural networks to learn the offsets of pseudo-LIDAR point clouds with inaccurate locations. We argue that the displacement of pseudo-LiDAR is not the natural form of objects, and thus learning the noisy representation of pseudo-LiDAR will lead to constant failures in real world problems. A learned refinement of pseudo-LiDAR is beneficial for a monocular 3D detection task. In our work, inspired by geometric constraints, we propose a novel depth refinement framework to predict object location, in which geometry information serves as a priori information. Another problem is that the inaccuracy of depth estimation at the edges of objects will lead to the long-tail problem in the depth distribution of foreground targets. Large amounts of noise will appear when we transform the estimated depth map into a pseudo-point cloud. Therefore, we try to limit the long-tail problem around objects by rightfully replacing the pseudo-LiDAR along the contours of objects in the depth map.

To solve the aforementioned problems, we propose a two-stage method to alleviate the 3D location error and the long-tail noise of objects. The advantages of our proposed method are two-folded. Specifically, by adopting geometry information in scene context as a constraint, we fuse object segmentation, depth estimation and the physical shapes of objects jointly as a priori information to refine the depths of foreground objects in our depth maps. Additionally, using the prior knowledge of objects’ physical shapes, we adjust the distribution of the pseudo-LiDAR point cloud accordingly. We propose a simple yet effective class-specific normalization strategy. Finally, a 3D object detection method based on point cloud is introduced into our pipeline to obtain 3D detection results.

A thorough evaluation of our proposed method was conducted. Both qualitative and quantitative results show that, by introducing geometric constraints as a priori information and fusing this prior information with object segmentation and depth estimation, the performance in predicting the depths of foreground objects can be significantly improved. The method can produce higher accuracy by 15% on the KITTI validation dataset than other methods. The class-specific normalization strategy is able to produce better pseudo-LiDAR point cloud representations, and led to better performance by 10% on the KITTI validation dataset, which includes small objects, such as pedestrians.

In summary, we propose a novel monocular 3D object detection framework that can improve the robustness of the pseudo-LiDAR-based method with the input of a single image. Our main contributions are as follows:We found that the main limitations of monocular 3D detection are due to the inaccuracy of the target position and the uncertainty of the depth distribution of the foreground target. These two problems arise from inaccurate depth estimation.We first propose an innovative method based on joint image segmentation and geometric-constraint-based target-guided depth adjustment to predict the target depth and provide the depth prediction confidence measure. The accuracy of the predicted target depth in the depth map is improved.We utilize the prior target size and normalization strategy to tackle the long-tail noise problem in pseudo-LiDAR point clouds. The uncertainly of depth distribution is reduced.Thorough experimentation has been carried out with the KITTI dataset. With the two novel solutions, our proposed monocular 3D object detection framework outperforms various state-of-the-art methods.

The paper is structured as follows. The related studies are reviewed in Section 2. We focus on various monocular 3D object detection models. In addition, publicly available datasets that were generated to facilitate research are introduced. Section 3 describes our proposed deep learning framework. We evaluate our framework and compare its performance with those of state-of-the art methods. Section 4 presents the experimental results and comparative analysis. Finally, in Section 5, we draw the conclusions and outline future research directions.

## 2. Related Work

We group the related works into three parts: monocular 3D object detection algorithms, monocular depth estimation methods and benchmark datasets. Due to the progress in deep neural networks, we focus on deep learning models in this section.

### 2.1. Monocular 3D Object Detection

Previous research utilized multimodal features. Besides the higher cost of the sensing equipment, the frameworks are complex. Ku et al. [12] proposed a 3D object detector (AVOD) that fuses features extracted from LiDAR and RGB images. To simplify the design for real-time autonomous driving applications, they adopted the same network structure for both feature extractors with fewer layers. Peng et al. [13] first used LiDAR point clouds to train a 3D object detector. In the second step, the LiDAR-based detector was further trained with the input of the RGB image.

Recent studies attempted to detect 3D objects without the use of 3D sensors. The advantages of monocular 3D object detectors are that system cost is reduced, and with the input of a single image, only one feature extractor is needed. To tackle the missing dimension, a straightforward idea is to directly add a regression head based on well developed 2D detection models to predict depth information from 2D images with a supervised learning strategy. Vajgl et al. [14] extended You Only Look Once (YOLO) to 3D detection by updating the original loss function with a part responsible for distance estimation. Mauri et al. [15] proposed a Lightweight CNN (L-CNN) method for real-time 3D object detection based on the YOLOv5 2D object detector. To improve the regression accuracy of depth information, Xie et al. [16] developed a novel depth-based stratification structure to improve depth predictions by establishing mathematical models for the relation between depth and image size for objects. Xiao et al. [17] adopted temporal information to improve the depth estimation accuracy for 3D detection. Another recent method of monocular 3D detection is to have a 2D image mimic a 3D LiDAR signal and deal with 3D detection task in 3D space. Weng and Kitani [8] converted a 2D image to the 3D representation, called a pseudo-LiDAR point cloud, via monocular depth estimation. This, with the aid of the 2D proposal estimated from the image, was then input to the LiDAR-based segmenter. Wang et al. [6] adopted the same idea. Their proposed pipeline first estimates the depth map from monocular or stereo images. 3D coordinates of pixels are back-projected into the 3D space to generate the pseudo-LiDAR point cloud. Experimentation was performed with two 3D object detection algorithms: the LiDAR-based method AVOD [12] and th e RGB-D-based method Frustum PointNet (F-PointNet) [18]. Ma et al. [7] proposed a two-stage framework. In 3D data generation stage, 2D proposals and a depth map are estimated from the RGB image. The data, represented as a pseudo-LiDAR point cloud, are input to the second stage of 3D box estimation based on F-PointNet [18].

### 2.2. Monocular Depth Estimation

The aforementioned studies regress the depth map of the monocular image and predict the distances of objects. To infer depth from 2D image is a challenging problem in computer vision. Most recent studies used a deep learning approach. Khan et al. [19] presented a comprehensive survey of this research topic. They grouped 13 state-of-the-art monocular depth estimation (MDE) methods into three categories: supervised, self-supervised and semi-supervised. Fu et al. [5] proposed a regression network (DORN) which comprises two major modules—a dense feature extractor and a scene understanding module. As there are no reliable cues, sp the accuracy of the depth map inferred from the 2D image is usually low. In order to improve the predicted distance, Shi et al. [10] proposed a novel geometry-based method to estimate the distance based on the physical height and the projected height of the object in the image. Similarly, Lu et al. [11] proposed GUPNet for monocular 3D object detection with the use of geometry priors (height parameters) for depth inference. Lian et al. [20] proposed a CNN for joint 3D object detection and depth estimation from images. To train the network, they devised a function to measure projected consistency loss based on the transformation relationship between target size and depth. Li et al. [21] utilized the relationships between neighboring objects and the relationships among all objects in a scene for MDE. The method can produce accurate depth maps for indoor scenes. However, it is difficult to model the object–object and object–scene relationships in images of outdoor scenes. In order to extract better features for depth inference, Liu et al. [22] proposed an encoder–decoder structured CNN with attention mechanisms. Xu et al. [23] also adopted the spatial attention for depth estimation. The proposed network works effectively, even on small targets. However, the training and inference processes are computationally demanding.

### 2.3. Benchmark Dataset and Evaluation Metrics

Benchmark datasets have been created to facilitate the research in 3D object detection. The KITTI dataset [9,24] contains records of traffic scenarios acquired by grayscale cameras, RGB cameras and 3D laser scanners. Annotations (3D bounding boxes) of targets (e.g., cars, pedestrians and cyclists) are widely used in 3D object detection research. A monocular 3D object detector can utilize a LiDAR-based 3D object detection algorithm pre-trained with LiDAR data. Zamanakos et al. [25] presented a comprehensive survey of LiDAR-based methods and three publicly available datasets (KITTI [24], and two newly released datasets, nuScenes [26] and Waymo Open Dataset (WOD) [27]). One common evaluation metric is the mean average precision (mAP) which is used along with a threshold on the intersection over union (IoU). However, mAP does not reflect the accuracy of 3D box orientation. KITTI proposed another metric—average orientation similarity (AOS). NuScenes proposed the nuScenes detection score (NDS), which is a combination of mAP and mean true positive (mTP) metrics. WOD modified the calculation of AP and introduced a new metric, average precision heading (APH) which measures the heading accuracy of target. Li et al. [28] also presented a recent review on benchmark datasets and evaluation metrics. Other datasets, e.g., Objectron [29] and Cityscapes [30], although containing only RGB images, also provide annotations of 3D objects that can be used for monocular 3D object detection research.

The monocular depth estimation module can be trained with some of the aforementioned datasets, e.g., KITTI [24]. Some publicly available datasets were created mainly for research on image-based depth inference; e.g., NYU-v2 [31] comprises indoor images, and Make3D [32] contains outdoor, indoor and synthetic scenes. Commonly used evaluation metrics involve the deviations of predicted depths from ground truth, e.g., root mean square error (RMSE) and absolute relative difference. Khan et al. [19] presented a review of the datasets and evaluation metrics for that specific problem. They [33] also presented a more recent and comprehensive review. The depth datasets are categorized in accordance with their intended applications.

## 3. Proposed Framework

An overview of our proposed framework is shown in Figure 3. A single RGB image is fed into two separate deep learning models: the monocular depth estimation network and the 2D instance segmentation network. The coarse depth map is improved by the depth distribution adjustment (DDA) module. On the other hand, the depth map and mask proposal are used to refine the depth information of instance segmentation with the geometric constraint depth refinement (GCDR) module. Inspired by GUPNet [11], we specifically replace the 2D bounding box priors with the 2D segmentation. The long-tail problem in the depth data is solved by reducing the influence of large depth error on the boundary of the object. With the depth data enhanced by these two modules and the camera matrix, a better 3D representation (pseudo-LiDAR point cloud) is generated. Finally, we use an off-the-shelf LiDAR-based 3D detection algorithm to infer the 3D detection results (location, scale and bounding box). The following sub-sections describe 2D instance segmentation, monocular depth estimation, depth refinement, pseudo-LiDAR point cloud generation and 3D object detection, in detail.

### 3.1. 2D Instance Segmentation

Pseudo-LiDAR-based monocular 3D detection tends to suffer from the problem of low accuracy. This is because, unlike real LiDAR data, pseudo-LiDAR data are generated from the depth map, which is estimated from a single RGB image. The estimated depth map has pixel-level density, and the information is insufficient. As the 2D object proposal serves as a prior, its estimation needs to be robust in order to improve the accuracy of 3D detection. Within the 2D bounding box in a depth map, there are redundant points between the boundaries of object and box. LiDAR-based 3D detection algorithms usually perform point cloud segmentation. The redundant points will certainly confuse the point cloud segmentation and affect the 3D detection. To remove the interference of redundant points, we use 2D instance segmentation instead of a 2D bounding box. We compare the point clouds converted from the refined depth map using 2D segmentation and a 2D bounding box in Figure 4. As displayed in the figure, by refining all the points using our proposed method within the 2D bounding box, redundant points lying between boundaries of the bounding box and segmentation are also dragged back, mistakenly, into the object’s 3D location. With 2D segmentation, the redundant points not enclosed by the ground truth location are largely removed. The use of 2D instance segmentation will result in a better pseudo-LiDAR point cloud for the subsequent 3D object detection. Specifically, we use the anchor-free CenterMask [34] as our instance segmentation network. It produces more accurate and robust 2D instance segmentation. Moreover, it is more capable of tackling partially occluded objects.

### 3.2. Monocular Depth Estimation

To convert the input RGB image into 3D space, a monocular depth estimation algorithm can provide the distance information for 3D object detection. We use the state-of-the-art monocular depth estimation algorithm DORN [5] as a sub-network in our framework. This pre-trained network is utilized as an offline module and integrated with other parts of our framework. Since our proposed framework is agnostic to different depth estimation algorithms, one can replace DORN with any other algorithms, if necessary, without affecting the final results.

### 3.3. Object-Guided Depth Refinement

Figure 5 shows the architecture of the proposed object-guided depth refinement module. Aiming to guide the model to refine the depth estimation only for foreground objects, we combine the pre-computed depth map and 2D instance segmentation to generate region of interest (RoI) features according to each predicted 2D object. We follow the geometry projection law of pinhole camera, in which the physical height of the object is needed. We modify and re-train CenterMask [34] by adding a sub-head to regress the 3D height with uncertainty with a Laplacian distribution for each object accordingly. To cope with the problems that both depth estimation and monocular 3D detection tend to struggle with when the objects are far away, we propose a penalty factor generated from the discretization of the depth distribution in order to optimize the learning process of our model, and to better predict the depth information of distant objects.

Directly regressing depth from a 2D detector is a hard and yet unreasonable problem. Inspired by GUPNet [11], we propose a geometric constraint depth refinement (GCDR) module to not only refine the depth of near objects, but also optimize the depth estimation of distant objects. Instead of avoiding the acquisition of depth from the 2D image, as in [11], we simply learn the depth uncertainty in a probability framework by geometry projection based on the assumptions from the depth map and 2D instance segmentation. In this way, we can enhance the depth refinement reliability.

As a prerequisite, a prediction of 3D height for each object is generated by adding a sub-head to the CenterMask [34] base model. We assume that the 3D height has uncertainty in a Laplacian distribution, as we simply do not fully trust the reliability of this information. The probability density function of a Laplace random variable χ∼La(μ,β) is $f\left(x\right)=\frac{1}{2\beta }e^{\frac{-|x-\mu |}{\beta }}$, where μ and β are the location and the diversity parameter. Thus, the mean and deviation of such a distribution are $\mu =\mu_{H},\sigma =\sqrt{2}\beta$. We re-train the CenterMask [34] model to predict the mean $\mu_{H}$ and the standard deviation $\sigma_{H}$, which represent the direct regression 3D height and the uncertainty of it, respectively.

To better predict distant objects, which is the bottleneck of current monocular depth estimation and 3D detection methods, we introduce a distance-sensitive factor (DSF) γ as a penalty to force the network to focus on faraway objects. We adopt a depth discretization strategy named linear interpolate discretization (LID) [10], as shown in Figure 6. The depth values are quantized in a way that the further the object, the larger the range of the depth bin the object may fall in, which is specifically calculated as follows:
(1)$$d_{i}=d_{min}+\frac{d_{max}-d_{min}}{D(D+1)}\cdot i\left(i+1\right)$$
where $d_{i}$ is the continuous depth value, $\left[d_{min},d_{max}\right]$ is the full depth range of an object, *D* is the number of depth bins to be separated and *i* is the depth bin index. Using this discretization strategy, the network can be more tolerant with the depth error and the DSF can be smoother. The DSF is calculated as follows:(2)$$\gamma =\log(d_{m})$$
where $d_{m}$ is the mean depth value in the depth bin in which the object is located. The exact depth of the object is calculated by adding half of the empirical object length and the mean depth at its 2D mask center. The loss function used to train the model to generate 3D physical height of each object is as follows:(3)$$Loss_{H}=\frac{\sqrt{2}}{\sigma_{H}}\left|\mu_{H}-H_{GT}\right|+\log\left(\gamma *\sigma_{H}\right)$$
where $H_{GT}$ is the ground truth physical height of the object, $\mu_{H}$ is the raw output of 3D regression head, and $\sigma_{H}$ is the uncertainty of the result. With the introduction of DSF, we may successfully alleviate the low confidence caused by the long distances of faraway objects, leaving the large $\sigma_{H}$ to indicate only the noisy labels or hard objects. With the prediction distribution of 3D height *H*, the depth distribution can be approximated using the geometry projection function:(4)$$D_{pred}=\frac{f\cdot H}{h}=\frac{f\cdot (\beta_{H}\cdot X+\mu_{H})}{h}=\frac{f\cdot \beta_{H}}{h}\cdot X+\frac{f\cdot \mu_{H}}{h}$$
where *h* is the projected height of the object in image, *f* is the focal length of the camera, $\mu_{H},\beta_{H}$ are the parameters of the predicted Laplace distribution, $La(\mu_{H},\beta_{H})$ is the 3D physical height and *X* is the standard Laplace distribution La(0,1). An additional bias generated from the pre-computed depth map is used to initialize the projection results. Its mean value $\mu_{bias}$ is also computed by combining the mean depth of the 2D mask center $C_{mask}$ and half of the empirical object length *L* as follows:(5)$$\mu_{bias}=C_{mask}+\frac{L}{2}$$

$\sigma_{bias}$ is the learned standard deviation multiplied by γ. Then, the final Laplace distribution of predicted depth *d* is computed as:(6)$$d=La\left(\mu_{pred},\sigma_{pred}\right)+La\left(\mu_{bias},\sigma_{bias}\right)$$
where
(7)$$\mu_{pred}=\frac{f\cdot \mu_{H}}{h}$$
(8)$$\beta_{pred}=\frac{f\cdot \beta_{H}}{h}$$
(9)$$\mu_{d}=\mu_{pred}+\mu_{bias}$$
(10)$$\sigma_{d}=\sqrt{{\sigma_{pred}}^{2}+{\sigma_{bias}}^{2}}$$

The final uncertainty of depth distribution contains both the projection uncertainty and the learned bias uncertainty, which can be optimized in our GCDR module using the depth refinement loss:(11)$$L_{depth\_refine}=\frac{\sqrt{2}}{\sigma_{d}}\left|\mu_{d}-d_{GT}\right|+\log\left(\gamma *\sigma_{d}\right)$$
where $d_{GT}$ is the ground truth depth value. Note that all the calculations are optimized also by γ, which allows the proposed method to be agnostic with respect to the distance of object.

GCDR aims to optimize depth accuracy in the depth map, as we crave the potential of using a 2D image to predict depth information in a geometric manner. For the convenience of our further adjustments in the depth map, a confidence score is necessary to indicate whether the result is trustworthy. Since our final distance non-sensitive standard deviation has the capability of indicating the uncertainty of depth without considering the distance of the object, we further project its value into the 0∼1 space via an exponential function to generate the depth confidence score:(12)$$S_{depth}=exp\left(-\sigma_{d}\right)$$
where $S_{depth}$ represents the weights. We treat the object depth $d_{m}$ in the depth map and the geometric constraint refined depth *d* with different levels of emphasis. The final refined depth of the object is calculated as follows:(13)$$D_{final}=d_{m}\cdot \left(1-S_{depth}\right)+d\cdot S_{depth}$$

Finally, we adjust the depth information within the 2D segmentation of each object using the final refined depth, and thus finish the refinement. Our proposed method adopts the geometric projection to avoid direct depth regression from the 2D image as a compensation measure. It alleviates the influence of large distances, utilizes the potential of a single 2D image, and enhances the depth estimation of both near and faraway objects.

### 3.4. Depth Distribution Adjustment

GCDR mainly addresses the problem of objects’ misplacement in the depth dimension, by refining and moving back objects into their correct locations. Still, using a depth map to generate the pseudo-LiDAR point cloud still has the problem of distortion. The object will be stretched out far beyond its actual physical proportions, which is called the long-tail problem. It is mainly caused by inaccurate depth estimation blurring around the object boundary. The object boundaries in pseudo-LiDAR are crucial, as they represent the shape of the object.

In order to minimize the influence of wrongly estimated depth around the boundary of an object, a depth distribution adjustment (DDA) module is proposed to adjust points with weak depth estimation into their correct locations via the adjustment of the distribution of depth information within each object’s mask. We trust 2D segmentation, as a priori knowledge that can be utilized to determine precisely the pixels that belong to the object. By observing the discretization depth of each pixel within a 2D mask, we assume that the depth distribution of each object follows the Poisson distribution. To solve the long-tail problem is to narrow the distribution of pixels in the mask. First, we set the peak of the Poisson distribution $\mu_{\gamma }$ as our anchor for further adjustment. Then, using the target scale, as in the empirical length of the object accordingly as the standard deviation $\sigma_{g}$, we can map the Poisson distribution to a discrete Gaussian distribution with mean and standard deviation $\mu_{\gamma }$ and $\sigma_{g}$, respectively, as follows:(14)$$P\left(\mu_{\gamma },\sigma_{\gamma }\right)\to G\left(\mu_{\gamma },\sigma_{g}\right)$$

In this way, the distortion of depth around the boundary of each object can be well removed, providing better shape representation after converting the information to a pseudo-LiDAR point cloud.

### 3.5. Pseudo-LiDAR Generation

Pseudo-LiDAR-based 3D detection methods usually take advantage of well-developed LiDAR-based 3D detection methods. Thus, integrating information at hand and mimicking the LiDAR signal is essential. With the help of the camera calibration files provided by public datasets, such as KITTI [24] and nuScenes [26], we can easily transform depth maps into point clouds. Given pixel coordinates in 2D image space (u,v) and the estimated depth *d*, the 3D coordinates $\left(X_{c},Y_{c},Z_{c}\right)$ in the camera coordinate system can be computed as:(15)$$Z_{c}=d$$
(16)$$X_{c}=\frac{\left(u-c_{x}\right)Z_{c}}{f_{x}}$$
(17)$$Y_{c}=\frac{\left(v-c_{y}\right)Z_{c}}{f_{y}}$$
where $f_{x}$ and $f_{y}$ are the focal length of the camera in *x* and *y* axes, respectively, and $\left(c_{x},c_{y}\right)$ is the principal point. With the additional extrinsic matrix $R_{t}$, we can also acquire the 3D location of each point in the world coordinate system by:(18)$$C^{-1}{\left[X_{c},Y_{c},Z_{c}\right]}^{T}$$
where *C* is camera parameter calculated by intrinsic and extrinsic parameters. Parameters of the calculation are provided by the dataset accordingly, which will be shown in Section 4.

### 3.6. 3D Object Detection

The pseudo-LiDAR point cloud generated from the refined depth map is then fed into a LiDAR-based 3D detection algorithm for 3D bounding box estimation. Note that our framework does not require a specific 3D detection algorithm. In this study, we experimented with two different methods, F-PointNet [18] and PVRCNN [35]. For F-PointNet [18], we replaced the approach of adopting a 2D bounding box to generate a frustum proposal with a 2D instance segmentation frustum. As for PVRCNN [35], no adjustments were implemented.

## 4. Result and Discussion

We evaluated the performance of the proposed framework on the KITTI bird’s eye view (BEV) and 3D object detection benchmark [24]. Furthermore, we performed ablation studies of the main modules—GCDR and DDA. As demonstrated in the quantitative and visual results, our proposed framework can specifically refine the depth estimation and boost monocular 3D object detection. Moreover, our framework can achieve good performance using any off-the-shelf LiDAR-based 3D detectors.

### 4.1. System Setup

The KITTI 3D object detection benchmark [24] contains 7481 and 7518 images for training and testing, respectively. The KITTI test set is only for online benchmarking, as it has no publicly available ground truth labels. Therefore, similarly to [36], we trained and validated our framework by separating the training set into training and validation subsets with 3712 and 3769 images, respectively. For each (left) image, KITTI provides the corresponding 64-beam Velodyne LiDAR point cloud, 2D and 3D bounding boxes and the camera calibration matrices. To emphasize that our framework is purely working on monocular images, we did not use LiDAR signals or stereo images in the training, validation or testing processes. Only the camera calibration matrix was used for projecting the depth map to pseudo-LiDAR.

Our framework focuses on the BEV and 3D object detection. We report the overall results on the test set, and ablation studies on the validation set. Differently from several previous studies [37,38], besides evaluating the "car" category, we also focused on the "pedestrian" category. The evaluation metric was average precision (AP) with the IoU thresholds of 0.5 and 0.7. The AP obtained for BEV and 3D object detection tasks are denoted as $AP_{BEV}$ and $AP_{3D}$, respectively. For each object category, KITTI separates all the samples into three degrees of difficulty (easy, moderate and hard) by thresholding the object’s 2D box height or occlusion/truncation level. The easier settings contain all the objects in the harder settings.

We compare our framework with several state-of-the-art image-based and pseudo-LiDAR-based monocular 3D detection methods. To be fair and show our superiority, the results of pseudo-LiDAR-based 3D detectors were generated from the same depth estimation algorithm. For some baseline methods with both monocular and stereo image versions, we only selected the monocular image version for comparison.

### 4.2. Implementation Details

The 2D instance segmentation produces an object mask, which is then fed into the GCDR module. A better mask will help to focus the refinement process on the relevant pixels in the depth map. We adopted the extended architecture VoVNetV2-99 of CenterMask, and added another regression head right after the feature pyramid network (FPN) to generate an estimated 3D height of the object. We froze the pre-trained checkpoint of VoVNetV2-99 and trained the 3D height regression head separately on the combined KITTI training set (training subset and validation subset).

We used an off-the-shelf monocular depth estimation method DORN [5] as our depth estimation base model. DORN was trained on 23,488 KITTI images, and some of these images may have overlapped with a popular split of the KITTI validation subset. For evaluation, we decided not to re-train the DORN model for two reasons. One is that most of the previous studies using DORN as a depth estimation sub-model have not re-trained it. For fair comparison, we decided to follow the same approach. The other reason is that the overlapped images are only in the validation subset. They would not influence the overall performance in the KITTI test set. We just used the validation subset for ablation studies.

To generate a pseudo-LiDAR point cloud using our refined depth maps, we followed the settings in [6]. The generation was done by back-projecting the depth map into the Velodyne LiDAR coordinate system using the camera calibration matrix provided by KITTI and neglecting points with heights larger than 1 m.

To prove that our GCDR can serve as a plug-in module in any of the state-of-the-art LiDAR-based 3D detectors, we used two different LiDAR-based 3D detectors—F-PointNet [18], which uses both LiDAR points and monocular images, and PVRCNN [35], which uses LiDAR points only. See Section 3.6. For F-PointNet, we specifically applied F-PointNet V1. Both models were trained from scratch on the 3712-image KITTI training subset for ablation studies, and then evaluated on the validation subset. Finally, the models were re-trained on the training set with 7480 images, and then evaluated on the official KITTI test set.

### 4.3. Quantitative Results

As shown in Table 1, we compare our proposed framework with other state-of-the-art monocular 3D detectors on the KITTI test set. Following the argument from GUPNet [11], it is possible to increase the confidence of depth information for better 3D detection by learning the geometry uncertainty. The overall results prove that the same plain idea is suitable for the pseudo-LiDAR. Our framework outperformed GUPNet by 5.1%, 3.05% and 1.76% on easy, moderate and hard settings of 3D detection, respectively. Moreover, our proposed framework outperformed Mono-PLiDAR [8] by large margins of 14.45%, 9.75% and 7.43% for easy, moderate and hard settings of 3D detection, though the latter leverages pseudo-LiDAR-format 3D detection. This comparison shows that with more accurate depth information at hand by projecting a 2D input into 3D space and adopting the well-developed LiDAR-based detector, better 3D object detection can be achieved. Additionally, with the use of a monocular image as input, a pseudo-LiDAR point cloud is still a suitable 3D representation.

Table 2 and Table 3 summarize the results of our proposed framework and other reference methods on the KITTI validation subset. For better comparison, we included $AP_{BEV}$ and $AP_{3D}$ results under different IoU thresholds. It can be seen that our proposed framework achieved superior results on the "car" category by large margins. Specifically, under the same conditions, our proposed framework outperformed the second best method AM3D [7] by 12.37%, 5.61% and 5.34% on the easy, moderate and hard settings, respectively. The significant improvements confirm our argument that the most important element that restricts the accuracy is the mislocation of objects in 3D space, especially the error of depth information. By refining the depth information using object-guided geometric constraint, the overall $AP_{BEV}$ can be improved by roughly the same degree as $AP_{3D}$.

When test our framework on the KITTI validation subset, we also found an interesting phenomenon related to pedestrian detection. Thus, we additionally report the pedestrian and cyclist detection AP in Table 4 and Table 5. Other monocular 3D detection methods suffer from poor performance in pedestrian detection and cyclist detection due to the small object sizes and small amount of training samples (2207 for pedestrians, 734 for cyclists and 14,357 for cars). Classes are heavily imbalanced. Specifically, our proposed method outperformed the best pseudo-LiDAR-based method in pedestrian detection. Our proposed framework was only slightly inferior to the reference method [8] on the hard setting of cyclist detection. The main reasons for this boost in performance in both pedestrian and cyclist detection were the contributions of the proposed GCDR and DDA modules. They provide more distilled information, and thus enable our proposed framework to better detect small-sized objects. As to the hard setting for cyclist detection, the poor performance was mainly due to the 2D detection failure.

### 4.4. Ablation Study

The GCDR module can boost the performance of 3D detection. The following parts are essential: geometry uncertainty learning (GUL) and geometric constraints (GC) via projection of physical height. To carefully evaluate each part, we first composed a pseudo-LiDAR based 3D detector directly using projection from depth map without any further procedure as a baseline model, which we called vanilla PL. Then, we compared it with two variant models. The first one adds a general geometric constraint (vanilla PL + GC)—to refine the depth information from the depth map uwing only the physical height of object. The other model is composed by adding geometry uncertainty learning only (vanilla PC + GUL). In this way, we can investigate the impacts of the two essential parts. As shown in Table 6, GC can deal with some of the distant objects by generating relatively correct depths for their locations, and thus outperformed the baseline model by 4.1%, 3.1% and 1.6% on the easy, moderate and hard settings, respectively. With GUL, GCDR further gains the ability to continuously learn the distribution of the depth information of objects with different distances. By doing so, the learning strategy can produce a more accurate depth for each foreground object. As a result, GUL outperformed the baseline model by 9.9%, 4.9% and 4.2% on the easy, moderate and hard settings, respectively. We can conclude that 3D object location is a vital element for monocular 3D object detection.

Next, we replaced the 2D instance segmentation with the 2D detector (Faster-RCNN [50]). 2D instance segmentation can benefit the overall pipeline and leverage the depth of the object in the right way. The pixels around the boundary of each object, which are fed into GCDR for further processing, are more precise. A 2D mask can eliminate the background pixels and guide GCDR module to only refine the foreground pixels to their rightful locations. The following projected pseudo-LiDAR will be well separated from background points, which leads to a better 3D detection result. As shown in Table 7, 2D instance segmentation outperformed 2D detection by 2.1%, 0.2% and 2.3% on the easy, moderate and hard settings, respectively.

We now compare our model with or without the proposed distance-sensitive factor (DSF) to deal with faraway objects. As shown in Table 8, by adding the DSF, our proposed model had better accuracy of car detection in moderate and hard settings by 2.3% and 2.4%, respectively, and retained the accuracy for easy samples. The reason for these results is that most of the faraway objects are in the moderate and hard settings, which our DSF specifically targets. Therefore, adding the DSF can significantly boost the performance of a monocular 3D detector when dealing with distant objects, where depth information cannot be directly sampled naturally.

We also quantified the contribution of the DDA module. It relies on the 2D instance segmentation. Therefore, the ablation study on the DDA module was performed based on the 2D instance segmentation. As shown in Table 9, our model with DDA can boost the performance of car detection by 6.5%, 3.3% and 2.0%, on the easy, moderate and hard settings, respectively. The other two categories are not obviously influenced by the DDA. The intrinsic cause of this phenomenon is two-fold—correct locations and poor distribution of pedestrians in the depth map. Pedestrians usually possess fewer pixels but are empirically suitable for 2D detectors. Cyclists suffer from the problems of wrong depth information and fewer training samples. On the contrary, cars have more training samples and allow better depth estimations and depth distributions around boundaries. Regardless of the different impacts, the DDA module can tackle the long-tail problem generated by the boundary effect of depth estimation and improve the overall performance.

Finally, to demonstrate the front end of our proposed framework can serve as a plug-in module to any LiDAR-based 3D detectors, we carefully selected two representative LiDAR-based 3D detectors, PV-RCNN [35] and F-PointNet [18], as our baselines. F-PointNet uses a 2D detector to generate frustum proposals for further training and evaluation. On the other hand, PV-RCNN generates proposals directly from 3D space using the voxelized method. As shown in Table 10, regardless of treating the 3D space as a frustum or voxels, or generating proposals from 2D detectors or directly from a point cloud, after adding our front end modules, the overall performances were roughly the same. We conclude that whatever the LiDAR-based detector, our modules can provide a good pseudo-LiDAR point cloud, and the complete monocular 3D object detection framework can produce reasonable accurate results.

### 4.5. Visual Results

We examine the visual results and further investigate the effectiveness of each part of our proposed framework here. As shown in Figure 7, with our GCDR and DDA modules, the pseudo-LiDAR point cloud generated from the raw depth map is well organized and contains more accurate depth information. The refined pseudo-LiDAR point cloud exhibits richer and more representative shapes and distributions, which are more suitable for a LiDAR-based 3D detector. For the sake of fairness, we used the same 3D detection method, F-PointNet [18]. We visualize the results from both frontal-view images (top row) and BEV point cloud maps (bottom row). Ground truth boxes are in green, and prediction results are in yellow. It can be seen that the refined pseudo-LiDAR point cloud was improved with more accurate object location and natural distribution after adding the GCDR module and DDA module. Therefore, the performance of pseudo-LiDAR based monocular 3D detection was improved.

Figure 8 shows the three results of car detection in complicated images. Each image contains cars at both near and far distances. Moreover, some cars are partially occluded. The top row shows the frontal-view images, and the bottom row shows the BEV point cloud maps. Ground truth boxes are in green, and prediction results are in yellow. We can observe that our proposed framework can produce remarkably accurate 3D boxes for occluded targets. Additionally, our framework can detect targets that are located far away. Note that our method still suffers from some bad results in certain situations, as shown in the middle column of Figure 8. Our method detected one false positive car. The reason for that is that our method generates results from refined depth maps with 2D segmentation masks. An occluded target will lead to a poor mask or failed segmentation. In that case, the point clouds within the mask area may be refined wrongly, and the overall result will be incorrect.

Figure 9 shows two results for the other two classes—pedestrian (left column) and cyclist (right column). In the left column, ground truth is marked in green and pedestrian detection result is marked in blue. In the right column, ground truths are marked in blue, and car and cyclist detection results are marked in green and yellow, respectively. The top row shows the frontal-view images, and the bottom row shows the BEV point cloud maps. View angle has been adjusted for better visualization. We can observe that our proposed framework can detect remarkably accurate 3D boxes for pedestrians and cyclists, and also our DDA module can reshape the divergent point cloud with a reasonable boundary.

## 5. Conclusions

Detection of 3D objects from a single 2D image is a challenging task. Generally, the predicted target location is not very accurate, and the depth data of the target are noisy. These two problems arise from the inaccurate depth estimation. In this paper, we proposed a monocular 3D object detection framework with two innovative solutions. First, we proposed the GCDR module to predict the target depth and provide the depth prediction confidence measure. With the integration of 2D instance segmentation and the coarse depth map, the predicted target location is improved. Second, we proposed the DDA module. By the use of the target scale information, DDA can adjust the depth data distribution and reduce the long-tail noise in the point cloud. With the refined depth information, the 3D representation of the scene, called the pseudo-LiDAR point cloud, is generated. Finally, we use a LiDAR-based algorithm to detect the 3D target. We conducted extensive experiments to investigate the significance of the two proposed modules. We evaluated the performance of the proposed framework on the KITTI dataset. Our proposed framework outperformed various state-of-the-art methods by more than 12.37% and 5.34% on the easy and hard settings of the KITTI validation subset, respectively. For the KITTI test set, our framework outperformed other methods by more than 5.1% and 1.76% on the easy and hard settings, respectively.

In our future work, we would like to conduct more experiments on other datasets, such as nuScenes [26]. Although our proposed framework outperforms many state-of-the-art monocular 3D object detection methods, there is still room for improvement. Since our method relies on the quality of 2D masks, and generating masks for partially occluded objects remains an unsolved problem, training an occlusion-aware segmentation method may be benefit for our pipeline. Another direction for improving monocular 3D detection may be trying to improve the inference strategy of depth prediction. As humans, we can infer the distance of one object if there is a known object in view. If we can train a model to infer object-related distance with the depth information of only a few learned objects, then its monocular depth estimation and 3D detection may be improved.

## Figures and Tables

**Figure 1 sensors-22-02576-f001:**
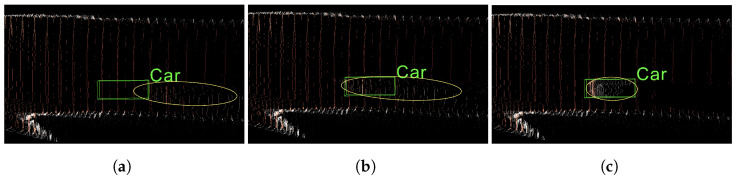
(**a**) Pseudo-LiDAR point cloud estimated by DORN [5] and ground truth; (**b**) pseudo-LiDAR point cloud estimated by our proposed depth refinement; (**c**) pseudo-LiDAR point cloud estimated by our proposed depth distribution adjustment. (Yellow ellipses indicate the object’s point cloud).

**Figure 2 sensors-22-02576-f002:**
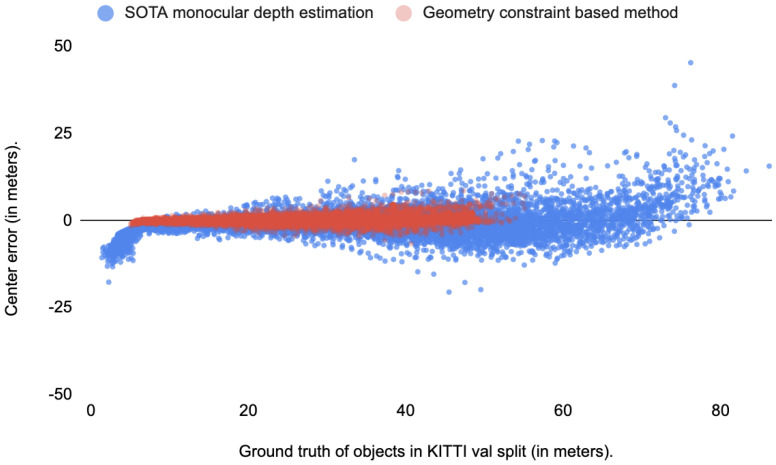
Comparison of monocular depth estimation errors between DORN [5] (blue data) and GUPNet [11] (red data).

**Figure 3 sensors-22-02576-f003:**
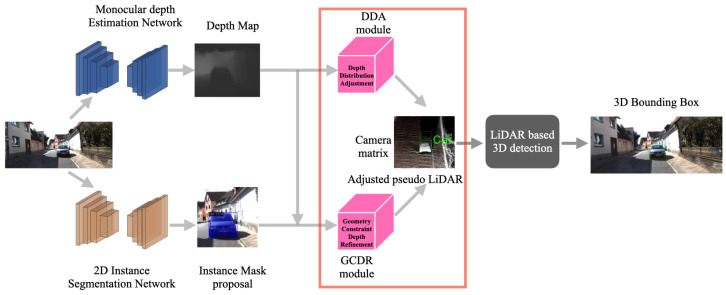
An overview of the proposed monocular 3D object detection framework.

**Figure 4 sensors-22-02576-f004:**
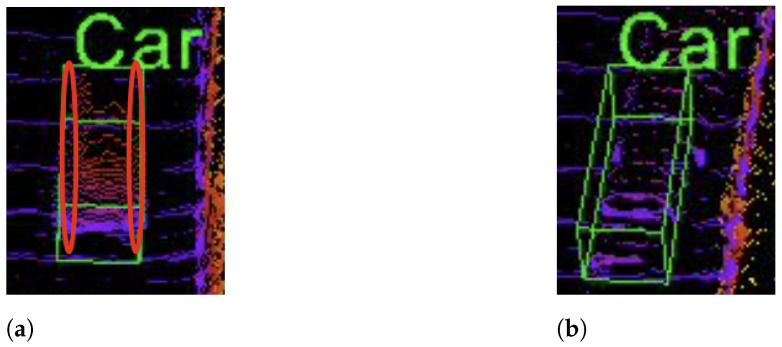
Comparison of 3D detection using: (**a**) a 2D bounding box; (**b**) 2D instance segmentation. Ellipses in Figure 4a denote the noise point clouds lying between the boundary and the bounding box of a certain object.

**Figure 5 sensors-22-02576-f005:**
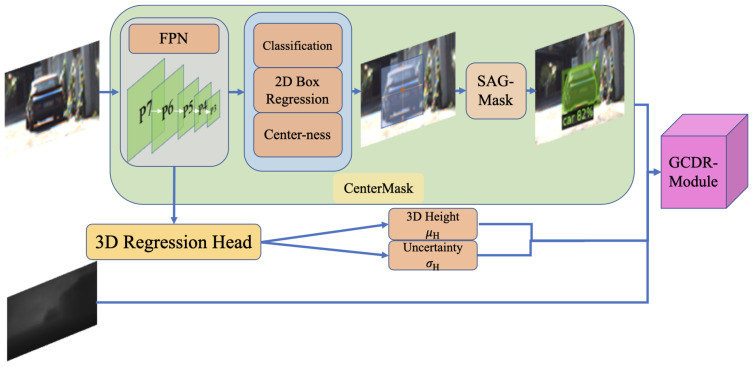
The complete object-guided depth refinement module. Our module takes an image as input, along with the original depth map, 3D height and uncertainty generated from the added 3D regression head. All is feed into our GCDR module for further depth refinement.

**Figure 6 sensors-22-02576-f006:**
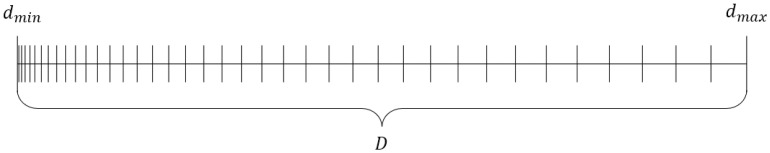
Linear interpolation discretization.

**Figure 7 sensors-22-02576-f007:**
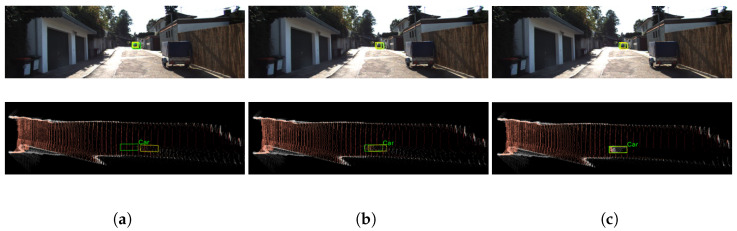
Results of car detection using: (**a**) vanilla PL; (**b**) vanilla PL + GCDR; (**c**) vanilla PL + GCDR + DDA. Zoom in for details. The same 3D detection method F-Pointnet [18] was adopted for comparison. Top row images are frontal-view images, and bottom row images are bird’s-eye-view point cloud maps. Ground truth boxes are in green, and prediction results are in yellow.

**Figure 8 sensors-22-02576-f008:**
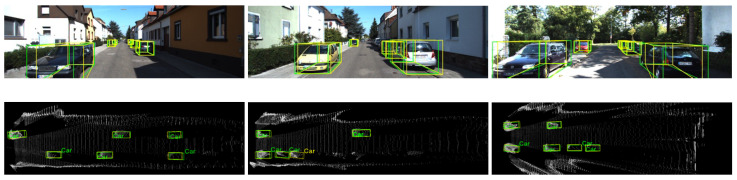
Detection of cars at different distances with occlusion. Top row images are frontal-view images, and bottom row images are bird’s-eye-view point cloud maps. Ground truth boxes are in green and prediction results are in yellow. Our proposed framework is able to create accurate 3D boxes at different distances and for occluded targets.

**Figure 9 sensors-22-02576-f009:**
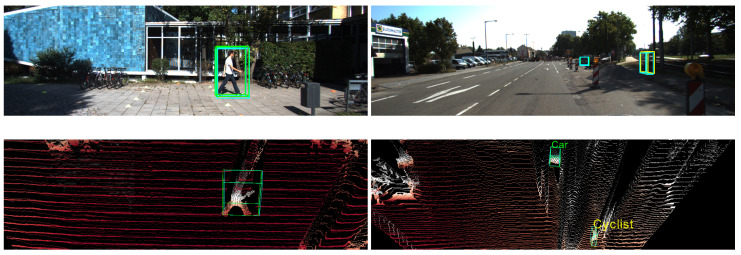
Detection of a pedestrian and a cyclist. Top row images are frontal-view, and bottom row images are bird’s-eye-view point cloud maps. Ground truth boxes are in green, and predicted results are in yellow. Our proposed framework is able to create accurate 3D boxes for a pedestrian and a cyclist, and our DDA module can produce reasonable shapes for point clouds.

**Table 1 sensors-22-02576-t001:** Performance of 3D detection of the "car" category on the KITTI test set. We indicate the highest results with red.

Method	Input	${\mathrm{AP}}_{3D}@\mathrm{IOU}=0.7$			${\mathrm{AP}}_{\mathrm{BEV}}@\mathrm{IOU}=0.7$		
		Easy	Moderate	Hard	Easy	Moderate	Hard
M3D-RPN [39]	Mono	14.76	9.71	7.42	21.02	13.67	10.23
AM3D [7]	P-LiDAR	16.50	10.74	9.52	25.03	17.32	14.91
RTM3D [40]	Mono	14.41	10.34	8.77	19.17	14.20	11.99
MonoPair [41]	Mono	13.04	9.99	8.65	19.28	14.83	12.89
SMOKE [42]	Mono	14.03	9.76	7.84	20.83	14.49	12.75
MoVi-3D [43]	Mono	15.19	10.90	9.26	22.76	17.03	14.85
D4LCN [44]	Mono	16.65	11.72	9.51	22.51	16.02	12.55
CaDNN [45]	Mono	19.17	13.41	11.46	27.94	18.91	17.19
PatchNet [46]	Mono	15.68	11.12	10.17	22.97	16.86	14.97
Mono-PLiDAR [8]	P-LiDAR	10.76	7.50	6.10	21.27	13.92	11.25
GUP Net [11]	Mono	20.11	14.20	11.77	30.29	21.19	18.20
Proposed framework	P-LiDAR	25.21	17.25	13.53	37.86	27.92	21.97
Improvement	−	+5.1	+3.05	+1.76	+7.57	+6.73	+3.77

**Table 2 sensors-22-02576-t002:** Performance of 3D detection of the “car” category under AP@IOU=0.7 of the KITTI validation subset. We indicate the highest results with red.

Method	${\mathrm{AP}}_{3D}@\mathrm{IOU}=0.7$			${\mathrm{AP}}_{\mathrm{BEV}}@\mathrm{IOU}=0.7$		
	Easy	Moderate	Hard	Easy	Moderate	Hard
Multi-Fusion [47]	10.53	5.69	5.39	22.03	13.63	11.60
Deep3DBox [48]	5.85	4.10	3.84	9.99	7.71	5.30
MonoGRNet [49]	13.88	10.19	7.62	19.72	12.81	10.15
M3D-RPN [39]	14.53	11.07	8.65	20.85	15.62	11.88
MonoPair [41]	16.28	12.30	10.42	24.12	18.17	15.76
GUPNet [11]	22.76	16.46	13.72	31.07	22.94	19.75
Mono-PLiDAR [8]	28.2	18.5	16.4	40.6	26.3	22.9
AM3D [7]	32.23	21.09	17.26	43.75	28.39	23.87
Proposed framework	44.6	26.7	22.6	56.6	33.4	29.4
Improvement	+12.37	+5.61	+5.34	+12.85	+5.01	+5.53

**Table 3 sensors-22-02576-t003:** Performance of 3D detection of the "car" category under AP@IOU=0.5 of the KITTI validation subset. We indicate the highest results with red.

Method	${\mathrm{AP}}_{3D}@\mathrm{IOU}=0.5$			${\mathrm{AP}}_{\mathrm{BEV}}@\mathrm{IOU}=0.5$		
	Easy	Moderate	Hard	Easy	Moderate	Hard
Multi-Fusion [47]	47.88	29.48	26.44	55.02	36.73	31.27
Deep3DBox [48]	27.04	20.55	15.88	30.02	23.77	18.83
MonoGRNet [49]	47.59	32.28	25.50	48.53	35.94	28.59
M3D-RPN [39]	48.53	35.94	28.59	53.35	39.60	31.76
MonoPair [41]	55.38	42.39	37.99	61.06	47.63	41.92
GUPNet [11]	57.62	42.33	37.59	61.78	47.06	40.88
Mono-PLiDAR [8]	66.3	42.3	38.5	70.8	49.4	42.7
AM3D [7]	68.86	49.19	42.24	72.64	51.82	44.21
Proposed framework	75.6	57.1	49.8	78.7	59.2	51.0
Improvement	+6.74	+7.91	+7.56	+6.06	+7.38	+6.79

**Table 4 sensors-22-02576-t004:** Performance of 3D detection of the “pedestrian” category of the KITTI validation subset. We indicate the highest results with red.

Method	${\mathrm{Pedestrain}}_{3D}@\mathrm{IOU}=0.5$			${\mathrm{Pedestrain}}_{\mathrm{BEV}}@\mathrm{IOU}=0.5$		
	Easy	Moderate	Hard	Easy	Moderate	Hard
Mono-PLiDAR [8]	11.6	11.2	10.9	14.4	13.8	12.0
Proposed framework	18.29	15.36	14.70	23.75	20.10	17.58
Improvement	+6.69	4.16	+3.80	+9.35	+6.30	+5.58

**Table 5 sensors-22-02576-t005:** Performance of 3D detection of the “cyclist” category of the KITTI validation subset. We indicate the highest results with red.

Method	${\mathrm{Cyclist}}_{3D}@\mathrm{IOU}=0.5$			${\mathrm{Cyclist}}_{\mathrm{BEV}}@\mathrm{IOU}=0.5$		
	Easy	Moderate	Hard	Easy	Moderate	Hard
Mono-PLiDAR [8]	8.5	6.5	6.5	11.0	7.7	6.8
Proposed framework	9.1	6.7	6.2	12.1	8.9	7.0
Improvement	+0.6	+0.2	−0.3	+1.1	+1.2	+0.2

**Table 6 sensors-22-02576-t006:** Ablation study on input data with the “car” category of the KITTI validation subset.

Method	${\mathrm{Car}}_{{\mathrm{AP}}_{3D}}@\mathrm{IOU}=0.7$			${\mathrm{Car}}_{{\mathrm{AP}}_{\mathrm{BEV}}}@\mathrm{IOU}=0.7$		
	Easy	Moderate	Hard	Easy	Moderate	Hard
Vanilla PL	28.2	18.5	16.4	40.6	26.3	22.9
Vanilla PL + GC	32.3	21.6	18.0	44.2	29.8	24.3
Vanilla PL + GUL	38.1	23.4	20.6	50.1	31.6	27.4

**Table 7 sensors-22-02576-t007:** Ablation study on object segmentation with the “car” category of the KITTI validation subset.

Baseline	${\mathrm{Car}}_{3D}@\mathrm{IOU}=0.7$			${\mathrm{Car}}_{\mathrm{BEV}}@\mathrm{IOU}=0.7$		
	Easy	Moderate	Hard	Easy	Moderate	Hard
Faster-RCNN [50]	40.0	23.2	18.3	49.7	29.6	24.5
CenterMask [34]	42.1	23.4	20.6	50.1	31.6	27.4

**Table 8 sensors-22-02576-t008:** Ablation study on distance-sensitive factor with the “car” category of the KITTI validation subset.

DSF	${\mathrm{Car}}_{3D}@\mathrm{IOU}=0.7$			${\mathrm{Car}}_{\mathrm{BEV}}@\mathrm{IOU}=0.7$		
	Easy	Moderate	Hard	Easy	Moderate	Hard
×	44.1	24.4	20.2	55.2	31.6	27.8
✓	44.6	26.7	22.6	56.6	33.4	29.4

**Table 9 sensors-22-02576-t009:** Ablation study on depth distribution adjustment with the “car” category of the KITTI validation subset.

Adjust Distribution	${\mathrm{Car}}_{3D}@\mathrm{IOU}=0.7$			${\mathrm{Car}}_{\mathrm{BEV}}@\mathrm{IOU}=0.7$		
	Easy	Moderate	Hard	Easy	Moderate	Hard
×	38.1	23.4	20.6	50.1	31.6	27.4
✓	44.6	26.7	22.6	56.6	33.4	29.4

**Table 10 sensors-22-02576-t010:** Ablation study on LiDAR-based detector with the “car” category of the KITTI validation subset.

3D Detection Method	${\mathrm{Car}}_{3D}@\mathrm{IOU}=0.7$			${\mathrm{Car}}_{\mathrm{BEV}}@\mathrm{IOU}=0.7$		
	Easy	Moderate	Hard	Easy	Moderate	Hard
F-Pointnet [18]	44.6	26.7	22.6	56.6	33.4	29.4
PV-RCNN [35]	45.2	27.1	22.8	58.0	34.2	30.3

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
