# Peer review of "Deep Learning-Based Monocular 3D Object Detection with Refinement of Depth Information"

_sensors, 2022, doi:10.3390/s22072576_

Round 1

Reviewer 1 Report

As mentioned by the authors recently many new papers appeared on 3D object detection. The authors present a nice literature overview and state of the art.

But recently new papers appeared which should be discussed

Xidong Peng et.all. Side: Center-based stereo 3D Detector with structure-aware instance depth information. arXiv:2108.09663v2

Zhouzhen Xie et all. A multi-scaledepth stratification based monocular 3D Object detection algorithm arXiv:2201.04341v1

Marek Vajgl  , Petr Hurtik  and Tomáš Nejezchleba . Dist-YOLO: Fast Object Detection with Distance Estimation. Appl. Sci. 2022, 12, 1354. https://doi.org/10.3390/app12031354

  1. Decoux1 et all. Lightweight convolutional neural network for real-time 3D object detection in road and railway environments. Journal of Real-Time Image Processing (2022)

Peng Xiao,1,2 Fei Yan,1 Jiannan Chi,2,3 and Zhiliang Wang. Real-Time 3D Pedestrian Tracking with Monocular Camera. Wireless Communications and Mobile Computing / 2022 /

Reviewer 2 Report

  1. This manuscript is well prepared. The theoretical parts have been checked to be solid and the validation parts are also adequately presented. I listed only a few comments below hoping to further improve the integrity of it.
  2. Page 1, line 10: “uncertainly” should be “uncertainty”.
  3. Page 10, Eq.(16): Xc should be Yc.
  4. Similarly, Eq(15-16) were not to establish the coordinates (Xc, Yc, Zc). It was more than to build up the relationships between Xc to Zc and Yc to Zc. So this part may need to be rephrased.
  5. The improvements shown in the last rows of Table 1-3 did not fit the numbers presented in the Abstract and Conclusion Sections.
  6. I would suggest authors to fix the font sizes of all the tables in page 13-15.
  7. Page 16: Figure 7-9 need more detailed captions. Also, I found no labels in the subfigure (a), (b) and (c).
